# Protein Biomarkers for the Diagnosis of Alzheimer’s Disease at Different Stages of Neurodegeneration

**DOI:** 10.3390/ijms21186749

**Published:** 2020-09-15

**Authors:** Mar Pérez, Félix Hernández, Jesús Avila

**Affiliations:** 1Departamento de Anatomía Histología y Neurociencia, Facultad de Medicina UAM, 28029 Madrid, Spain; mar.perez@uam.es; 2Centro de Biología Molecular Severo Ochoa (CSIC-UAM), 28049 Madrid, Spain; fhernandez@cbm.csic.es; 3Network Center for Biomedical Research in Neurodegenerative Diseases (CIBERNED), 28031 Madrid, Spain

**Keywords:** tau, amyloid, PET, CSF

## Abstract

Mainly obtained from familial Alzheimer’s disease patients’ data, we know that some features of the neurodegenerative start several years before the appearance of clinical symptoms. In this brief review, we comment on some molecular and cellular markers appearing at different stages of the disease, before or once the clinical symptoms are evident. These markers are present in biological fluids or could be identified by image techniques. The combined use of molecular and cellular markers will be of interest to determine the development of the different phases of the disease.

## 1. Introduction

Tissue degeneration occurs during aging. Early markers to detect this process are needed to prevent such degeneration. In this regard, Alzheimer’s disease (AD), one of the most prevalent neurodegenerative conditions, is characterized by two main aberrant structures, namely senile plaques (extracellular deposits composed of beta-amyloid (Aβ) peptide oligomers) and neurofibrillary tangles (intraneuronal aggregates composed of tau protein).

There are two types of AD, one with a familial origin (FAD) and the other sporadic (SAD). In the former, plaques appear up to 20 years before clinical symptoms of the disease, whereas tangles occur around 5 years before [1]. The development of these clinical features takes place at different ages in FAD (early-onset) and SAD (late-onset). Developing before 60 years of age, FAD is caused by the presence of mutations at specific sites in three genes: APP, PSEN1 and PSEN2. These mutations are very early markers of the disease [2]. In contrast, the onset of SAD occurs later, after 65 years of age. Some genetic variants (not mutations) have been described in this form of AD. These genetic variants are risk factors for early onset of the disease. In this regard, variant 4 of ApoE accelerates the onset of the condition [3]. Additionally, aging is a major risk for the development of AD [4]. Moreover, several risk factors (see Table 1) related to lifestyle can also accelerate the development of dementia [5]. Life style factors, unlike aging or genetic factors, are modifiable factors and their modification could retard the development of the disease [5].

## 2. Results

### 2.1. AD Diagnosis

Autopsy criteria based on the presence of plaques and tangles in the brain for the proper and final diagnosis of AD were defined many years ago [10]. However, earlier diagnoses, based on early biomarkers, are required to prevent, or retard, the development of the disease. In this regard, the concentrations of components of plaques (amyloid) and tangles (tau) in fluids like cerebrospinal fluid (CSF) have been used as biomarkers of the disease [11], and image analysis, like positron emission tomography (PET), has been used to detect the presence of amyloid or tau aggregates [12].

### 2.2. CSF

CSF allows for the removal of waste products from the brain, among these the residues resulting from dead or damaged cells [13]. Measurement of changes in the concentration of Aβ, tau or phosphorylated tau in CSF is now standard practice for the diagnosis of AD. Some patients show increased levels of tau and phosphorylated tau in CSF, due to its release from damaged or dying neurons, but reduced levels of amyloid, possibly because of the conversion of soluble amyloid into aggregates of the peptide, which do not circulate in CSF [14,15]. However, at different phases of the disease, different levels of total tau or different levels of phosphorylated tau, modified at specific residues present at specific regions of its molecule, could be found in the CSF of the patients. Since distinct regions of tau can be phosphorylated [16], we will comment on that.

### 2.3. Tau Protein Phosphorylation

Tau, from the central nervous system, can be modified by phosphorylation at different residues along its molecule tau is a phosphorylated protein, containing 85 potential serine (S), threonine (T) and tyrosine (Y) phosphorylation sites. Phosphorylation regulates many processes such as the binding to microtubules [17], its subcellular distribution and axonal transport [18,19]. The binding of tau to microtubules is regulated by its phosphorylation at specific sites of its molecule [16]. Tau phosphorylation, at different residues, correlates with the presence of different pathologies (tauopathies), including Alzheimer’s disease [20,21]. Aberrant phosphorylation causes the disassembly of microtubules and changes in the organization of the neuronal cytoskeleton. In addition, phosphorylated tau can accumulate at the neuron cytoplasm to induce the formation of tau oligomers and aggregates like fibrillar filaments and tangles [22]. Tau is a substrate for protein kinases, especially proline-directed kinases such as GSK3β [23] but cyclin dependent kinases and mitogen activated protein kinases also phosphorylate Tau [24,25]. GSK3β and cdk5 may play a relatively more prominent role in tau phosphorylation in the human brain [23].

Phosphorylation of different tau sites (Tyr18, Ser199, Ser202/Thr205, Thr231, Ser262, Ser396 and Ser422) could be used to measure the progression of Alzheimer’s disease [26], at the determinate by looking at postmortem issues. We will comment below about it.

Hyperphosphorylation of tau may result from an imbalance in the activity of tau protein kinases and tau phosphatases. Several studies have revealed the role of phosphatase PP2A on tau dephosphorylation [27,28]. Regulation of the phosphorylation state and microtubule-binding activity of Tau by protein phosphatase 2A [28]. Furthermore, PP2A inhibition produces in vivo deregulation of many brain Ser/Thr kinases implicated in AD, including GSK3β [29] and cdk5 [30]. These kinases are also known as tau kinase 1 and tau kinase 2, respectively, for example [17]. Thus, PP2A dysfunction can induce several events that contribute to neuronal and synaptic damage in AD. Several therapeutic strategies are based on the activation of this phosphatase [31].

Recently, a soluble phosphorylated tau signature that links tau, amyloid and the stages of FAD has been reported [32]. The presence of P-tau 217 and P-tau 181 correlates with the development of amyloid plaques at earlier stages of the disease, before neuronal atrophy and changes in brain metabolism occur. P-tau 217 over P-tau 181 have been proposed as early markers of the disease in plasma samples [33]. Furthermore, neuronal atrophy is accompanied by P-tau 205/P-tau 202. Later on, tau tangles develop and total tau increases in the brains of AD patients [32]. Several protein kinases can modify the phosphorylated tau residues. In this context, P-tau 217 and P-tau 181 are modified by protein kinase cdk5, which is activated by Aβ [34,35]. This activation can occur during very early (asymptomatic) stages of the disease. Afterwards, tau phosphorylation at site 205 (probably modified by GSK3) and at site 202 (modified by GSK3) [16] can occur, correlating with a decrease in gray matter (loss of dendritic arborization and synaptic transmission), yielding the stage of neuronal dysfunction observed in the AD brain [32]. Finally, the development of tau tangles, related to an increase in total tau in CSF, announces the appearance of clinical symptoms and the onset of AD. In addition to Aβ and tau, other markers such as neurofilament light chain NF-L and CX3CL1 (fractalkine) are found in the CSF of AD patients. The presence of NF-L, or other neurofilament proteins, in blood correlates with the axon damage that occurs during neurodegeneration. This damage causes the release of neurofilament proteins into CSF and these proteins are then circulated into the blood. Thus, quantitative analyses for detecting NF-L (or phospho neurofilament heavy chain (NF-H)) in CSF (or blood) have been developed for the diagnosis of neurodegenerative diseases [36]. On the other hand, damaged neurons may secrete lower amounts of CX3CL1 than healthy neurons. It may result in a change (activation) of microglia cells. In this regard, lower levels of CX3CL1 have been reported in the CSF of AD patients compared to that of non-demented subjects [37]. It is known that microglia could be activated by amyloid beta and it enhances inflammation [38], and facilitates the development of tau pathology [39]. Although, there are other possible AD markers related to microglia cells, little is known about their presence in CSF [40].

Furthermore, the presence of nucleic acids in CSF (or blood) as putative AD biomarkers has been put forward. A low concentration of mitochondrial DNA in CSF has emerged as a preclinical marker of AD [41]. Interestingly, this marker is not present in other tauopathies like Creutzfeldt–Jakob disease [42]. In addition, in blood cells, the methylation of specific genes can be found mainly in AD patients, thereby pointing to the utility of these epigenetic markers to detect preclinical stages of AD [43]. Additionally, some microRNAs in plasma samples from AD patients are increased compared to controls [44]. However, to date, the concentrations of tau and Aβ in plasma have been the most widely studied potential markers of AD. For Aβ, the characterization and levels of autoantibodies present in AD patients have been studied [45]. Regarding tau, an increased level of P-tau 181 has been observed in the plasma of AD patients [46], but not in other tauopathies [33]. In most recent publications, the presence of p-tau 217 in plasma from people at very early preclinical stages has been reported [47,48]. In general, the use of plasma rather than CSF analysis is increasing since the former involves non-invasive extraction of samples. Furthermore, the isolation of tears for the detection of possible AD biomarkers may provide an even less invasive approach. In this regard, a recent study has corroborated the presence of specific miRNAs in the tears of AD patients [49]. This finding is, therefore, a good starting for the search of potential AD markers in this fluid.

### 2.4. Positron Emission Tomography

Another diagnostic approach in AD is the analysis of the presence of Aβ or tau protein aggregates by positron emission tomography (PET). In this technique, radiopharmaceutical tracers, linked to a radioisotope, bind to the Aβ or tau aggregates. The presence of aggregates is detected by measuring radioisotope decay. Two radioisotopes, namely C^11^ or F^18^, are commonly used. C^11^ has a half-life of 20 min, whereas F^18^ has a half-life of 110 min, making the latter more suitable. Additionally, compounds related to thioflavin (ThT), which binds to β-sheet regions, have been used to bind to Aβ and tau aggregates. Tracers were first designed for the former, the Pittsburg B compound called PIB-C11 being a pioneer of these compounds. PET is a non-invasive tool for amyloid imaging in humans [50]. In addition, PIB-C^11^ shows stronger binding to Aβ oligomers than ThT [51]. Moreover, when using PET, PIB-C^11^ can be complemented by FDG-F^18^, a tracer of glucose metabolism, since an inverse relation between these signals has been reported in the brains of AD patients [52].

PIB-C^11^ and other compounds like flutemetanol F^18^, florbetapir F^18^ and florbetaben F^18^ react mainly with Aβ aggregates, whereas the tracers AZF469 or FDDNP can bind to both Aβ and tau [53,54].

It is known that looking at clinical symptoms, the disease could be divided in different stages, but to measure underlying pathology that is present previously to the appearance of clinical features, the use of amyloid PET could be a suitable tool [55]. Additionally, this tool could be used for differential diagnosis of neurodegenerative and neuroinflammatory disorders [56]. On the other hand, visual evaluation of images is the standard way used by Aβ PET examination data. Since that way could result in a subjective analysis, an objective quantitative evaluation for image interpretation has been suggested [57]. However, no main changes were obtained by the comparison of the PET-only amyloid quantification method and a visual evaluation, using the 14C-Pittsburg compound (PiB) for PET analysis. The use of other radioactive tracers mainly F18 tracers, like florbetapir, flutemetanol or florbetaben, have allowed one to find a good correlation between, for example, florbetapir images and cortical amyloid pathology or to differentiate areas with frequent plaques from those containing sparse plaques [58]. Additionally, PET analysis can be used to look for asymmetries of Aβ-fibrillary plaques burden, an asymmetry, that could be associated with microglia activation in AD or in mouse models, for the disease [59].

Several tracers have been described for PET imaging of tau. The first generation of tracers included FAV-1451-F^18^ THK-5317-F^18^, T807 and PBB33 and the second-generation MK-624-F^18^, RO-948 and JNJ 311-F^18^. Both first- and second-generation tau tracers are based on the detection of specific binding to the β-sheet regions of aggregated tau [52]. Various tau filament folds have been described for aggregates present in tauopathies like AD and cortical basal degeneration (CBD) [31]. These distinct folds indicate different β-sheet regions that react in a particular manner to the tracers.

A possible advantage of PET imaging for tau over amyloid for AD diagnosis is that the former predicts not only the amount but also the localization of the atrophy. This observation thus indicates that PET imaging for tau outperforms that for amyloid in the context of monitoring AD development [60].

### 2.5. How the Data of CSF and PET Could Complement Each Other to Indicate the Progress of the Disease

Based on the amyloid cascade hypothesis [61], amyloid pathology could facilitate tau pathology, for example by increasing tau phosphorylation. In this way, extracellular amyloid could bind to a cellular receptor, upon binding to the receptor, a signaling pathway could be activated and, as a consequence of that, different tau kinases could activate to modify specific sites at the tau molecule. Different amyloid receptors could bind to different receptors. At the very early stage, phosphotau 181 and 217 could be found in CSF [47]. At that stage, the level of amyloid is increased, and it can be already identified by PET analysis [32]. Amyloid, at that level, should bind to higher affinity receptors. An example could be the binding of amyloid to the alpha 7 nicotinic receptor [62]. Upon the binding cdk5 could be activated promoting the phosphorylation at tau residues 181 and 217. This modification may take place in some specific neurons that could be damaged and secrete phosphorylated tau to CSF. In addition, cdk5 (also known as tau kinase 2) facilitates amyloid processing of APP and cdk5 activator, p25 enhances cdk5 but reduced GSK3 (also known as tau kinase 1) activity [63].

At the later stages, the amyloid level increases and it could bind to other cell receptors, involving alternative signaling pathways and activation of other kinase like GSK3. For example, amyloid could bind to an insulin receptor [64,65], increasing GSK3 activity and in the phosphorylation of tau residues 202 and 205. Other suitable receptor could be the α_2A_ receptor, amyloid beta oligomers bind α_2A_ adrenergic receptors, this triggers GSK3 activation and it results in tau phosphorylation at residue 202 [66] (see Figure 1). Phosphorylation at different sites may occur in different neurons at different times. Neurons containing phosphorylated tau at specific residues could secrete that protein to the extracellular space and it could be found in CSF. Different phosphorylations can occur at different stages of the disease and could indicate different stages of neurodegeneration. Finally, the presence of tau tangles, which can be analyzed by PET studies, correlates with an increase in the amount of total tau in CSF [32]. Thus, CSF and PET analysis could complement each other to indicate the stage of Alzheimer’s disease.

## 3. Conclusions

In summary, in this short review, we briefly commented on the nature of some biomarkers obtained from fluid samples, mainly CSF and plasma of AD patients, and on AD diagnosis using imaging techniques, which are based on the detection of the two major hallmarks of AD, namely amyloid (senile plaques) and tau (neurofibrillary tangles) aggregates.

In addition to the use of the aforementioned biomarkers, additional tests should be conducted to obtain a complete picture of the status of a potential AD patient. These analyses may involve neurological observation, neuropsychological tests or the use of electroencephalography (EEG) or event-related potential (ERP) indices [67] as biomarkers of AD [68]. In this regard and the context of cognitive function, the contribution of gamma oscillations also emerges as a relevant feature to examine in potential patients [69]. Additionally, other biomarkers of cognitive deficiency, like the failure of the default mode network (DMN), have been proposed as earlier biomarkers of AD [70]. In this regard, magnetic resonance analyses are also important. Thus, the use of magnetoencephalography as a potential non-invasive technique to study brain function and to determine progression to dementia could be a relevant tool in the future [71]. Thus, these many factors (and markers) can serve to indicate the development of AD. In some cases, additional information may be required to determine how the disease will develop in a potential Alzheimer’s patient, as suggested in the Alzheimer Precision Medicine Initiative [72].

Although this review does not address tau-targeting therapies for AD, good references can be found in [73,74].

## Figures and Tables

**Figure 1 ijms-21-06749-f001:**
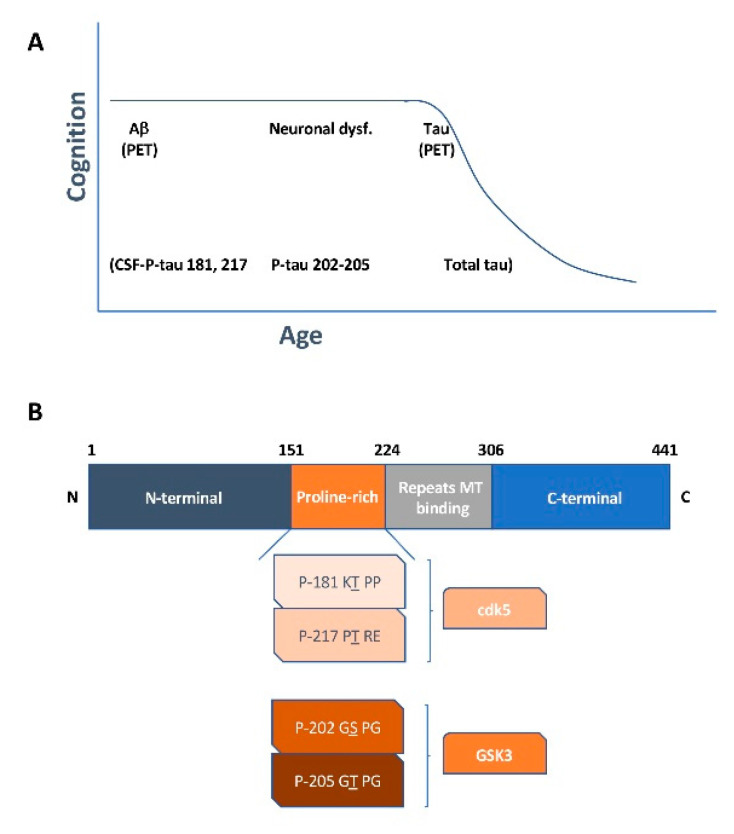
Phosphotau as the AD marker. (**A**) The diagram shows several markers for AD progression. Phosphorylation at several tau epitopes can be used as a marker for AD. At the very early stage, phosphotau 181 and 217 could be found in cerebrospinal fluid (CSF). At that stage, the level of amyloid is increased, and it can be already identified by PET analysis. At later stages, amyloid level increases and it could bind to other cell receptors, involving alternative signaling pathways and activation of other kinase like GSK3. (**B**) Schematic diagram of the Tau protein showing phosphorylation at residues, modified by GSK3 and cdk5, at the proline rich region of the molecule.

**Table 1 ijms-21-06749-t001:** List of risk factors to accelerate aging and neurodegenerative disorders like Alzheimer’s disease (AD). Several of these factors have been indicated [6,7] and some of them, like diabetes, obesity or hypertension, are risk factors for cardiovascular diseases [8]. On the other hand, chronic stress could be a main and earlier risk factor to accelerate aging [9].

Risk Factors to Accelerate Aging
Chronic infections
Physical inactivity
Dysbiosis
Diet
Chronic stress
Disturbed sleep
Diabetes
Obesity
Hypertension
Hypercholesterolemia

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
