# Peer review of "Protein Biomarkers for the Diagnosis of Alzheimer’s Disease at Different Stages of Neurodegeneration"

_ijms, 2020, doi:10.3390/ijms21186749_

Round 1

Reviewer 1 Report

It is well written, but I think the section on imaging needs significant expansion. There was no discussion on plague imaging the types of plagues etc. The manuscript is very general more like a perspective than a review

Author Response

Following his/her comments, we have done a further comment on amyloid plaques imaging.

Reviewer 2 Report

It is a short review of literature in this area,moderately orginal -trying to link imaging with blood biomarkers. It achieved what is possible in a short review. The conclusions consistent with the evidence and arguments presented. The text clear and easy to read, but English grammar needs correction in both abstract and through the manuscript. 

Author Response

As suggested by the reviewer, we have done a further English editing of the manuscript, by a native English speaking person.

Reviewer 3 Report

The review is named “Markers for the diagnosis of Alzheimer disease (AD) at different stages of neurodegeneration” and principally correctly describes some related data, mainly tau protein phosphorylation. However to be published the manuscript should be significantly modified. First, the title should be changed since only some protein biomarkers are described. Second, in addition to a very brief mentioning of PET mainly CSF proteins and CSF-PET complementation are described, although the current tendency is to switch to plasma/serum AD biomarkers (especially for screening and early diagnosis). More attention is devoted to tau phosphorylation but again most recent publications about plasma p-tau-217 are not quoted. (For example1-3)

Thus, to be published the manuscript needs some work.

----------------------------------------------------------------------------------

  1. Simon Dujardin, et al. Tau molecular diversity contributes to clinical heterogeneity in Alzheimer’s disease. Nature Medicine volume 26, pages 1256–1263(2020).
  2. Nicolas R Barthélemy et al. Blood plasma phosphorylated-tau isoforms track CNS change in Alzheimer's disease. J Exp Med. 2020; 217(11):e20200861.
  3. Palmqvist S. et al. Discriminative Accuracy of Plasma Phospho-tau217 for Alzheimer Disease vs Other Neurodegenerative Disorders. JAMA. 2020 Jul 28.

Author Response

  1. We have introduced a change in the title of the work, as proposed by the reviewer.
  2. We have comment further on plasma/serum AD biomarkers, focusing on recent publications on phosphorylated tau (p-tau217) found in plasma. References were also included.

Round 2

Reviewer 1 Report

The review is a little too short, more like a mini review or update

Line:330-331 rewrite,

Table 1 is a list, not a table

Does each factor carry the same risk?

No supporting references included in the list

Only one Figure included

Author Response

In this revised version, we have considered the points raised by reviewer 1, who has suggested a minor revision.

In this revised version, we have considered the points raised by reviewer 1, who has suggested a minor revision

  • Lines 330-331 of the text correspond to the list of references. Thus, the reviewer is probably indicating those lines that were present in the previous version. We believe that those lines correspond to current lines 201-203, and we have rewritten those lines. Please, confirm. In table 1, we have indicated that is, indeed, a list of modifiable risk factors for aging acceleration and for neurodegenerative disorders like AD.
  • Also, we comment, in the legend of the table (list) that chronic stress is a main modifiable risk factor.
  • Finally, new references for the point indicated about modifiable risk factors for aging or AD are enclosed.